# Acute stress reduces effortful prosocial behaviour

**Paul AG Forbes**[1]\*, **Gökhan Aydogan**[2], **Julia Braunstein**[1,3], **Boryana Todorova**[1], **Isabella C Wagner**[1,3,4], **Patricia L Lockwood**[5,6], **Matthew AJ Apps**[5,6], **Christian C Ruff**[2], **Claus Lamm**[1,3]\*

[1]Department of Cognition, Emotion, and Methods in Psychology, Faculty of Psychology, University of Vienna, Vienna, Austria; [2]Zurich Center for Neuroeconomics, Department of Economics, University of Zurich, Zurich, Switzerland; [3]Vienna Cognitive Science Hub, University of Vienna, Vienna, Austria; [4]Centre for Microbiology and Environmental Systems Science, University of Vienna, Vienna, Austria; [5]Centre for Human Brain Health, Institute of Mental Health and School of Psychology, University of Birmingham, Birmingham, United Kingdom; [6]Institute for Mental Health, School of Psychology, University of Birmingham, Birmingham, United Kingdom

**\*For correspondence:**
paul.forbes@univie.ac.at (PF);
claus.lamm@univie.ac.at (CL)

**Competing interest:** The authors declare that no competing interests exist.

**Abstract** Acute stress can change our cognition and emotions, but what specific consequences this has for human prosocial behaviour is unclear. Previous studies have mainly investigated prosociality with financial transfers in economic games and produced conflicting results. Yet a core feature of many types of prosocial behaviour is that they are effortful. We therefore examined how acute stress changes our willingness to exert effort that benefits others. Healthy male participants – half of whom were put under acute stress – made decisions whether to exert physical effort to gain money for themselves or another person. With this design, we could independently assess the effects of acute stress on prosocial, compared to self-benefitting, effortful behaviour. Compared to controls (n = 45), participants in the stress group (n = 46) chose to exert effort more often for self- than for other-benefitting rewards at a low level of effort. Additionally, the adverse effects of stress on prosocial effort were particularly pronounced in more selfish participants. Neuroimaging combined with computational modelling revealed a putative neural mechanism underlying these effects: more stressed participants showed increased activation to subjective value in the dorsal anterior cingulate cortex and anterior insula when they themselves could benefit from their exerted effort relative to when someone else could. By using an effort-based task that better approximates real-life prosocial behaviour and incorporating trait differences in prosocial tendencies, our study provides important insights into how acute stress affects prosociality and its associated neural mechanisms.

## eLife assessment

This study reports **useful** findings on the influence of acute stress on prosocial behavior and its neural correlates. The approach is **solid**, combining neuroimaging and neuroendocrine measures with computational cognitive modeling. The results will be of interest to researchers seeking to better characterize the influence of stress on neural computations mediating complex social behavior.

## Introduction

A perceived lack of resources to deal with the demands of the environment, such as during a tough job interview or a busy day at work, leads to the experience of acute stress. This can have profound effects on cognition and physiology with potential consequences for the way we behave towards

others (*Faber and Häusser, 2022*; *von Dawans et al., 2021*). However, previous work investigating prosocial decision-making under acute stress has yielded mixed findings. Some studies suggest that acute stress can increase prosocial tendencies (*Tomova et al., 2017*; *von Dawans et al., 2012*), whereas others have demonstrated decreases (*Sollberger et al., 2016*; *Vinkers et al., 2013*) or no effects at all (for a recent meta-analysis, see *Nitschke et al., 2022a*; *Veszteg et al., 2021*). Several mechanisms have been proposed as to how acute stress leads to increases in prosocial behaviour including an increased affiliative drive to seek social support from others: the 'tend and befriend' response (*Taylor et al., 2000*). Alternatively, decreased prosociality under stress could result from an increased self-focus (*Rimmele and Lobmaier, 2012*; *Tomova et al., 2014*) and/or changes in reward sensitivity (*Berghorst et al., 2013*).

Differences in prosocial decision-making under stress have typically been found in economic games requiring financial transfers to others, such as the dictator game or trust game (*Camerer, 2003*). These involve decisions about how to share a financial endowment with others but they do not involve much effort. Yet, prosocial behaviour in everyday life – from filling in a form to donate to charity to helping a colleague at work – requires the investment of differing amounts of effort (*Lockwood et al., 2017*; *Lockwood et al., 2021*; *Lockwood et al., 2022*). It is unclear how stress impacts this type of prosocial behaviour. Additionally, we do not know whether changes in sharing financial endowments reflect genuine differences in prosocial motivation under stress or differences in how stressed individuals value their own monetary gains (*Berghorst et al., 2013*; *Carvalheiro et al., 2021*; *Tomova et al., 2020*). Therefore, we directly compared the effects of acute stress on effortful *prosocial* behaviour compared to effortful *self-benefitting* behaviour. In the task, participants had to exert actual physical effort to obtain rewards either for themselves or another person. We then used computational modelling to investigate the neural mechanisms responsible for the potential stress-induced changes in effortful prosocial behaviour. Finally, we tested whether individual differences in existing prosocial tendencies modulated these effects.

Acute stress may reduce effortful prosocial behaviour via its more general effects on motivation and effort exertion. In rodents, acute stress results in a reduced willingness to exert effort for rewards without changing preferences for these rewards themselves (*Shafiei et al., 2012*) these effects are dependent on the hypothalamic–pituitary–adrenal (HPA) axis (*Bryce and Floresco, 2016*). In humans, acute stress also leads to an increased avoidance of tasks involving cognitive (*Bogdanov et al., 2021*) and physical (*Voulgaropoulou et al., 2022*) effort. However, previous work has focused solely on effort-based decisions for self-relevant outcomes. It is not known how acute stress affects effort-based decisions when someone else benefits from our effort, nor is it clear how acute stress affects the neurocomputational mechanisms of effort-based decision-making either for ourselves or for others.

We examined these mechanisms by focusing on how the computationally derived *subjective values* (SVs) of rewards are altered by changing the effort costs required to obtain them (*Bartra et al., 2013*; *Chong et al., 2017*; *Lockwood et al., 2017*; *Lockwood et al., 2021*; *Lockwood et al., 2022*). SV decreases as the effort costs required to obtain the reward increase. By using computational modelling, we could extract participant-specific effort discounting parameters (*K*) which represented the extent to which each participant devalued (or 'discounted') rewards by effort (*Lockwood et al., 2017*). That is, the discount parameters represent how much the SV of a reward decreases when a certain amount of effort has to be invested to obtain it. We used functional magnetic resonance imaging (fMRI) to establish brain regions where blood oxygen level-dependent (BOLD) activation responded to SV (*Lockwood et al., 2022*) and how this activation was affected by acute stress. We had a particular focus on the dorsal anterior cingulate cortex (dACC) and anterior insula (AI) as these areas are strongly implicated in effort-based decision-making (for a meta-analysis, see *Lopez-Gamundi et al., 2021*; *Pessiglione et al., 2018*). For instance, *Lockwood et al., 2022* found responses in these areas to SV on both self- and other-benefitting trials when participants were deciding whether to exert effort. In addition, both areas are sensitive to the effects of stress (*Berretz et al., 2021*; *Cerqueira et al., 2007*; *King et al., 2009*; *Kogler et al., 2015*; *Radley et al., 2005*). Thus, we hypothesised that if acute stress results in reduced effort-related prosocial behaviour (e.g. *Sollberger et al., 2016*; *Vinkers et al., 2013*), it may do so via a preferential response of these areas to SV related to self- compared to other-benefitting decisions in stressed participants. Conversely, if acute stress leads to increased effort-related prosocial behaviour (e.g. *Tomova et al., 2017*; *von Dawans et al., 2012*), dACC and AI should respond more to SV for other- compared to self-benefitting rewards in stressed participants.

The willingness to engage in prosocial behaviour shows substantial individual differences (*Murphy et al., 2011*; *Thielmann et al., 2020*) and may account for previous inconsistencies in terms of how acute stress affects social behaviour (*Nitschke et al., 2022a*). For example, *Azulay et al., 2022* showed that elevations in cortisol following stress induction resulted in greater generosity in a one-shot dictator game in participants scoring high on trait empathy, with such cortisol increases resulting in reduced generosity in those scoring low on empathy. Similarly, *Speer et al., 2022* showed that acute stress can accentuate existing differences in dishonesty (cf. *Schulreich et al., 2022*; *Ying et al., 2022*). Thus, we aimed to test whether individual differences in participants' trait prosocial tendencies, as measured by social value orientation (SVO; *Murphy et al., 2011*), which quantifies participants' tendency to distribute resources between themselves and another person, modulated the impact of acute stress on effortful prosocial behaviour. Specifically, we tested the hypothesis that individualistic participants (i.e. those who are more selfish) would show the most marked changes in effortful prosocial behaviour following acute stress.

## Results

### Experimental approach

Ninety-six male participants were randomly assigned to undergo an established stress induction protocol or a control task (*Dedovic et al., 2005*; *Tomova et al., 2017*). Saliva samples and perceived

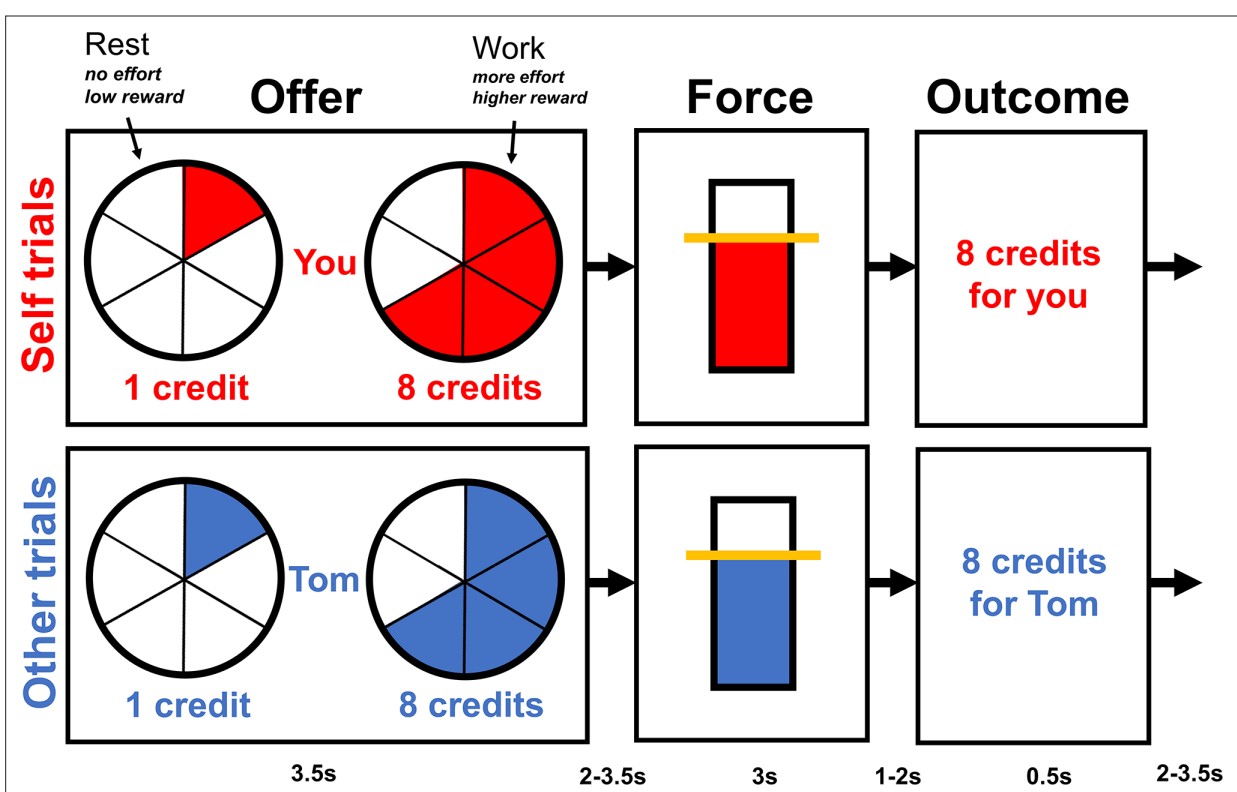

**Figure 1.** Overview of the task structure. During the offer phase, participants made choices between *rest*, a fixed low reward (1 credit) involving no effort, and *work*, a variable higher reward (2, 4, 6, 8, or 10 credits) involving more effort (30, 40, 50, 60, or 70% of participant's individual maximum voluntary contraction [MVC]). Higher effort levels were indicated by a more filled in circle. Participants had 3.2 s to make their choice, and the chosen option was then highlighted with a box for 0.3 s. If participants decided to work, then during the force phase, they had to squeeze the hand dynamometer at the required effort level (shown by the yellow line) for 1 s during a 3-s window. If they decided to rest, then the yellow line was displayed at the bottom of the bar and participants did not have to squeeze. During the outcome phase, the number of credits participants earned either for themselves or the next participant was displayed. Each phase of each trial was separated by a variable jitter shown at the bottom of the diagram.

The online version of this article includes the following figure supplement(s) for figure 1:

**Figure supplement 1.** The relative summed Bayesian information criterion (BIC) values for each model.

stress ratings were collected as indicators of physiological and subjective stress responses throughout the experiment. Participants made incentivised choices in an fMRI scanner between 'work' (exerting effort) or 'rest' (not exerting effort) (*Figure 1*; *Lockwood et al., 2021*; *Lockwood et al., 2017*; *Lockwood et al., 2022*). If they chose 'rest', participants waited until the end of the trial and received one credit; if they chose 'work', then they could receive a larger reward (2, 4, 6, 8, or 10 credits; converted to money at the end of the experiment). To receive this larger reward, participants had to squeeze a hand dynamometer for at least 1 s at the required effort level (30, 40, 50, 60, or 70% of their maximum voluntary contraction [MVC]) during a 3-s window. To establish whether stress affected effort-based decision-making differently for self-benefitting rewards compared to other-benefitting rewards, on half the trials, participants themselves were the recipients of the credits (self trials), whereas on the other half of trials participants were told the next participant in the study would receive the credits (other trials). As all participants were men, the name of the next participant was gender matched (all participants were told he was called Thomas; see 'Materials and methods'). Moreover, as participants did not see or interact with the next participant, familiarity was controlled across participants. We hypothesised that acute stress would decrease participants' willingness to exert other-benefitting prosocial effort compared to self-benefitting effort. Moreover, using neuroimaging combined with computational modelling, we examined whether the effects of stress were related to changes in SV representations in dACC and AI on self- compared to other-benefitting trials.

## Greater cortisol and perceived stress following stress induction

Participants rated their perceived stress at eight timepoints distributed across the experiment. We conducted a repeated-measures ANOVA with group as a between-subject factor and the eight sample timepoints as a within-subject factor. This revealed a significant interaction between group and sample timepoint, $F(7,644) = 19.05$, p<0.001, $\eta^2 = 0.101$, showing that the increase in stress over time was greater in the stress group than in the control group (*Figure 2*). Follow-up tests revealed a significant difference between the groups at timepoints 2–7 (all ps<0.04) but not at timepoints 1 or 8 (all p-values were Bonferroni corrected for multiple comparisons; for changes in other emotions during the experiment, see *Figure 2—figure supplement 1*). For salivary cortisol (*Figure 2*), we also found a significant interaction between group × sample timepoint, $F(5,445) = 6.945$, p<0.001, $\eta^2 = 0.028$. Follow-up tests revealed a significant difference between the groups at timepoints 3, 4, and 5 (all ps<0.001) but not at timepoints 1, 2, or 6 (all p-values were Bonferroni corrected). To create a measure of participants' total stress response throughout the experiment, we calculated the area under the curve (AUC) with respect to ground for participants' stress ratings and salivary cortisol (*Pruessner et al., 2003*).

## Acute stress reduces effortful prosocial behaviour at a low level of effort

To determine differences in choices to engage in effortful behaviour, we conducted a mixed-effects logistic regression using the *glmer* function from the R package *lme4* (*Bates et al., 2015*). Participants' choices were entered as the dependent variable (work = 1; rest = 0), with the factors Group (stress, control), Recipient (self, other), Effort (five levels), and Reward (five levels) as fixed effects. As random effects, we included random intercepts for participant and a random slope for Recipient (*Barr et al., 2013*). Models with a more complex random-effects structure (i.e. with Reward or Effort as random slopes) were singular or did not converge (despite using BOBYQA optimisation). A model containing all the three-way interactions between the fixed effects was compared to a more complex model containing also the four-way interaction using the *anova* function in R. The more complex model did not provide a better fit to the data (AIC: 7109.5 [four-way interaction model] vs. 7094.1 [three-way interactions model]; BIC: 7885.6 vs 7749.6; p=0.413). For the more parsimonious three-way interaction model, there was a significant interaction between Group, Recipient, and Effort (type III Wald test, $\chi^2[4] = 21.48$, p<0.001). We ran follow-up tests using the *emmeans* package in R (*Lenth, 2022*). Here, we tested the interaction between Group and Recipient at each level of effort, while keeping the reward level constant (average reward level). We found a significant Group × Recipient interaction at effort level 2 (p<0.05: Bonferroni corrected for multiple comparisons). Similarly, in an exploratory follow-up analysis, when we contrasted the two lower effort levels (effort levels 1 and 2) with the two higher effort levels (effort levels 4 and 5), we found an interaction between Group and Recipient at the

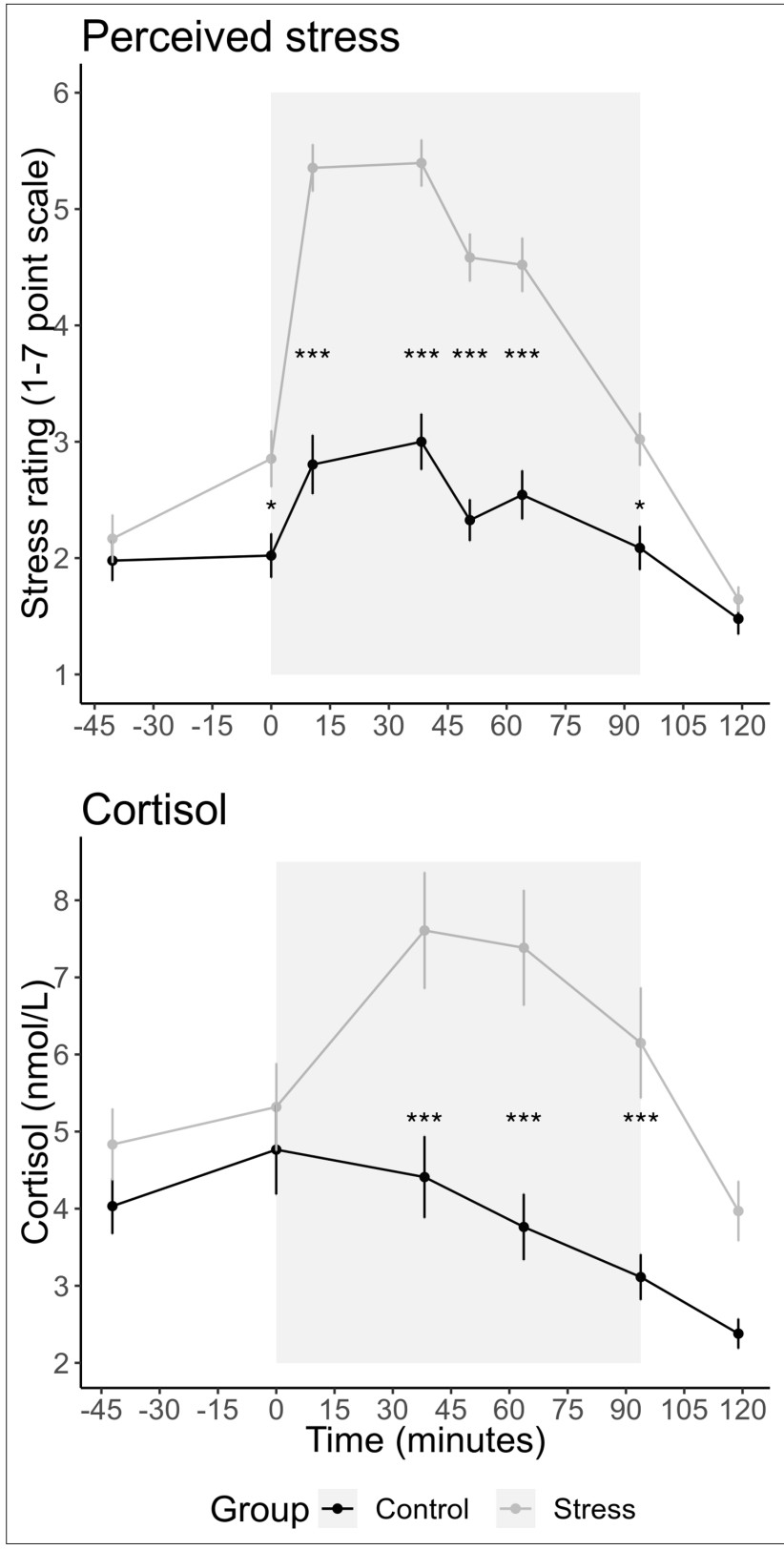

**Figure 2.** Mean (± SEM) salivary cortisol and stress ratings at each sample timepoint. The grey box indicates the time participants spent in the scanner doing the tasks. Participants completed six runs of 25 trials for the prosocial effort task. Before each run, participants experienced either an adapted version of the Montreal Imaging Stress Test (MIST; *Dedovic et al., 2005*) or the counting task from the Trier Social Stress Test (*Kirschbaum et al.,*

*Figure 2 continued on next page*

*Figure 2 continued*

1993). The asterisks indicate the significant level for the pairwise tests having corrected for multiple comparisons (*p<0.05; ***p<0.001).

The online version of this article includes the following figure supplement(s) for figure 2:

**Figure supplement 1.** Mean (± SEM) ratings at each sample timepoint for each emotion for the control group (solid line) and stress group (dashed line).

combined lower effort levels (p=0.035; Bonferroni corrected) but not at the combined higher effort levels (p=1.00).

This pattern of results was driven by the finding that whereas participants in the control group did not show a significant difference in the likelihood of deciding to exert effort on self trials compared to other trials at effort level 1 (p=0.595) or 2 (p=0.882), participants in the stress group favoured putting in effort on self trials compared to other trials even at effort level 1 (p<0.001) and 2 (p<0.001). Thus,

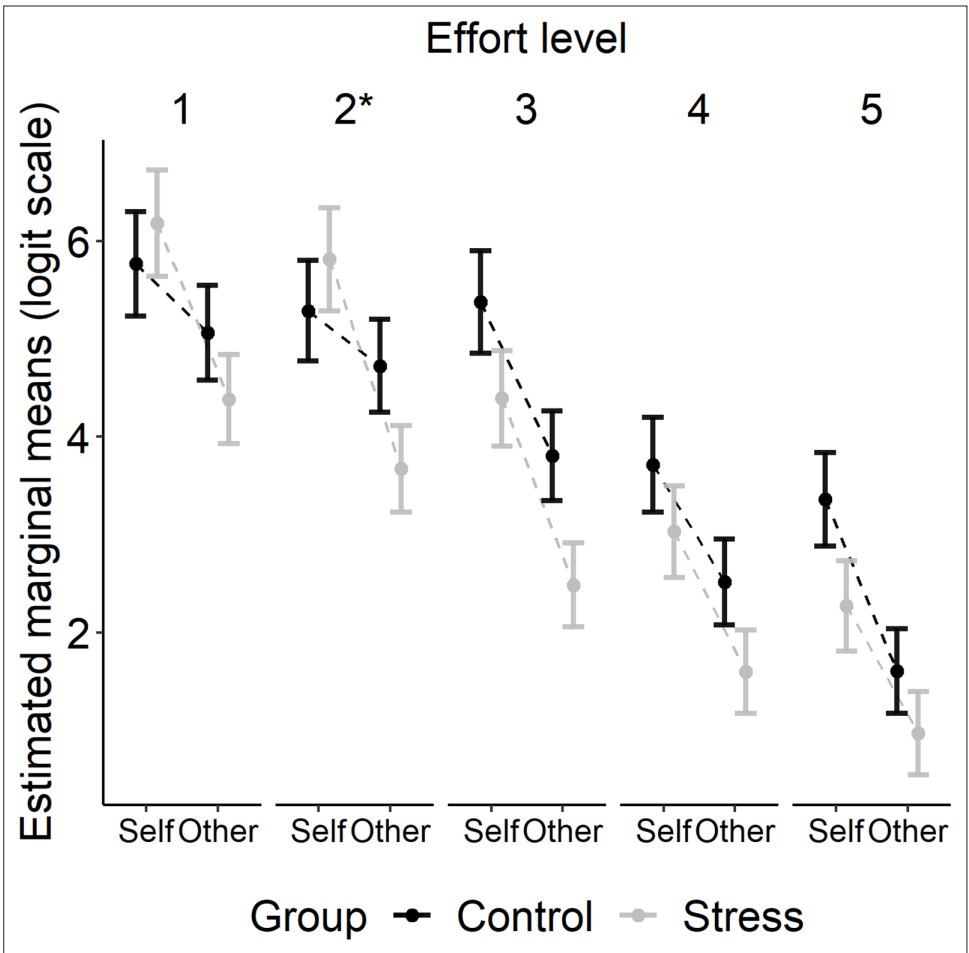

**Figure 3.** Estimated marginal means (± SEM) are plotted on a logit scale and were extracted using the *emmeans* package in R (*Lenth, 2022*). Follow-up tests showed that the three-way interaction between Group, Recipient, and Effort (type III Wald test, $\chi^2[4] = 21.48$, p<0.001) was driven by a significant interaction between Group and Recipient at effort level 2 (*p<0.05).

The online version of this article includes the following figure supplement(s) for figure 3:

**Figure supplement 1.** Immediately before and after the experiment, participants squeezed the hand dynamometer to each effort level and, on a 21-point Likert scale, rated (1) how much effort they exerted, (2) how physically demanding it was, and (3) how uncomfortable it was to squeeze to the required effort level.

**Figure supplement 2.** Estimated marginal means (± SEM) for each block of the prosocial effort task for the stress group and control group.

both our planned and exploratory analyses suggest that unlike participants in the control group, those in the stress group were less willing to put in effort to gain rewards for another person relative to themselves at a lower level of effort. The estimated marginal means for the three-way interaction between Group, Recipient, and Effort are shown in *Figure 3*.

There was a significant interaction between Group and Effort (type III Wald test, $\chi^2[4] = 17.27$, p=0.002), but follow-up tests did not reveal any significant differences at any effort level between the groups (all ps>0.26; Bonferroni corrected for multiple comparisons). There was a significant interaction between Recipient and Reward (type III Wald test, $\chi^2[4] = 19.12$, p<0.001) and also between Recipient, Effort, and Reward (type III Wald test, $\chi^2[16] = 50.65$, p<0.001) showing that the effect of Effort on Reward was different on self- compared to other-benefitting trials. The three-way interactions between Group × Recipient × Reward (p=0.285) and Group × Effort × Reward (p=0.064) were not significant (see *Supplementary file 1c and d* for the full model output of the choice data).

To analyse the force data (i.e. how much effort participants actually exerted once they decided to do so), we ran a mixed-effects model using the *lmer* function in R (*Bates et al., 2015*). The outcome variable was defined as the AUC during the duration of the force period relative to participants' MVC (i.e. the cumulated effort exerted over time). The model contained the same fixed effects as above and again random intercepts for participant and a random slope for Recipient. A type III Wald $\chi^2$ test revealed neither a main effect of Group nor any significant interaction involving Group (see *Supplementary file 1e* for the full model output) but a significant three-way interaction between Recipient, Effort, and Reward ($\chi^2[16] = 46.56$, p<0.001) and a two-way interaction between Effort and Reward ($\chi^2[16] = 67.92$, p<0.001).

Finally, if participants chose to put in effort, there were no significant differences between the groups regarding how often they successfully squeezed to the required effort level and received the reward (control group: mean success rate = 98.73%, SD = 2.19%; stress group: 97.66%, SD = 3.52%).

To summarise, at a lower level of effort, participants in the stress group showed a preference for exerting effort for self- compared to other-benefitting rewards. The control group, in contrast, shown no difference in their willingness to exert effort for a reward on self- compared to other-benefitting trials at this level of effort. There were no significant group differences in the amount of force exerted by the participants on each trial nor in how successful participants were in squeezing to the required effort level. This suggests that acute stress did not impact participants' ability to perform the effort task but that it did affect their decisions to put in effort for another person's benefit.

We conducted additional analyses to rule out the influence of potential fatigue and block effects. The stress group rated squeezing to the required effort level as more physically demanding immediately after the experiment compared to before, which was not seen in the control group (*Figure 3— figure supplement 1*). However, this was not related to the number of effortful choices for self or other rewards (*Supplementary file 1b*). Moreover, when we conducted the same mixed-effects logistic regression on participants' choices but also included the interaction between Group, Recipient, and Block, there was no significant three-way interaction between these factors, nor a significant two-way interaction between Group and Block (*Figure 3—figure supplement 2*). Additionally, the three-way interaction between Group, Recipient, and Effort was unaffected when controlling for potential block effects (type III Wald test $\chi^2[4] = 22.06$, p<0.001). Thus, whilst the stress group rated squeezing to the required effort level as more physically demanding following the experiment, this was not related to the number of effortful choices (for self or other) and the effects of Block on effortful choices (for self or other) did not differ between the group. Thus, changes in how physically demanding participants rated squeezing to the effort levels did not influence decisions to exert effort.

## Social value orientation modulates the impact of perceived stress on effortful prosocial behaviour

To test the hypothesis whether more individualistic participants become even more selfish under stress, we first calculated the proportion of effortful prosocial choices participants made relative to the total number of effortful choices they made (% prosocial choices). In other words, this measured how often participants chose to put in effort (i.e. 'work') for the other person and thereby gain a higher reward for the other person relative to the total number of times they chose to put in effort to gain a higher reward (i.e. both for themselves and the other person). This enabled us to determine each

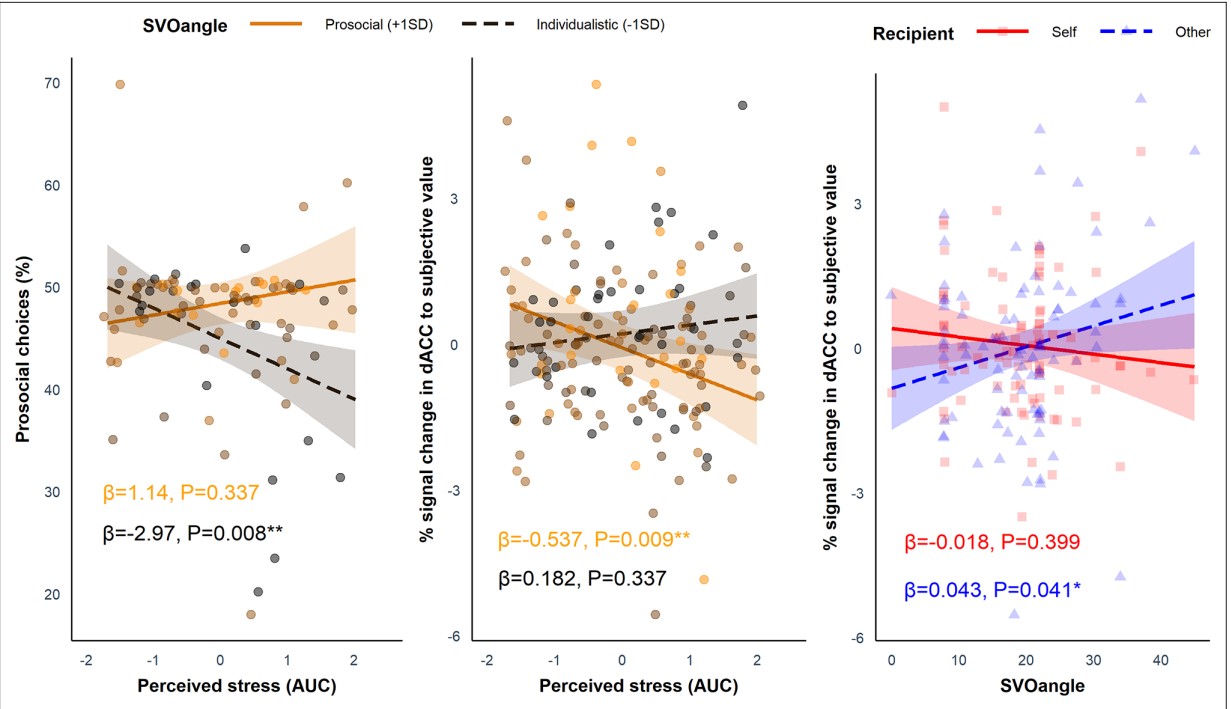

**Figure 4.** Simple slopes analyses showing the effects of perceived stress and SVO angle on behavioural and neural responses. **Left panel**: the interaction between social value orientation (SVO) angle and perceived stress for the proportion of prosocial choices (B = 0.246, SE = 0.103, p=0.020). Participants with a more individualistic SVO angle (–1 SD; black dashed line) became more selfish (reduced proportion of prosocial choices) at increasing levels of perceived stress; this was not seen in more prosocial participants (+1 SD; orange solid line). **Middle panel**: the interaction between SVO angle and perceived stress (B = −0.044, SE = 0.018, p=0.017) for dorsal anterior cingulate cortex (dACC) activation to subjective value (SV). Participants with a more prosocial SVO angle showed reduced activation in the dACC to SV (collapsed across self and other trials) at increasing levels of perceived stress; this was not seen in participants with a more individualistic SVO angle. **Right panel**: interaction between SVO angle and Recipient for dACC activation to SV (B = 0.061, SE = 0.028, p=0.034). Participants with a more prosocial SVO angle showed increased activation to SV$_{other}$ (blue dashed line). Responses to SV$_{self}$ (red solid line) did not change at increasing SVO angle. The ribbons represent the 95% confidence intervals and each point represents the individual data points from the participants (note, in the middle and right panel each participant provided two data points - one for the self condition and one of the other condition).

participant's prosocial effort relative to any general stress-induced decline in effortful behaviour. This measure combined all reward and effort levels.

There were no existing differences in SVO angle between the groups (control group mean = 19.33, SD = 8.67; stress group mean = 19.23, SD = 8.14; p=0.956). We found that across the whole sample – independent of the stress manipulation – there was a significant correlation between SVO angle and the proportion of prosocial choices (r = 0.225, p=0.032). So, as expected, those with a more prosocial SVO angle showed a higher proportion of prosocial choices in the task.

We tested whether SVO angle and Group (i.e. stress vs. control) interacted to influence the proportion of prosocial choices using linear regression in R but did not find a significant interaction (B = 0.164, SE = 0.181, p=0.368). Despite average differences between the groups in AUC for perceived stress and cortisol, there was considerable heterogeneity in the way participants responded to the stress induction. Thus, using linear regression in R, we investigated whether AUC for perceived stress (z scored across participants), which captured the total stress response of each participant in both the stress group and the control group throughout the experiment, interacted with SVO angle to influence the proportion of prosocial choices. Here, we found a significant interaction between SVO angle (range = 0–45°) and perceived stress (B = 0.246, SE = 0.103, p=0.020). Simple slopes analysis revealed that those with a more individualistic SVO angle (–1SD; 10.92°) showed a significant decline in the proportion of prosocial choices at increasing levels of perceived stress (B = −2.97, SE = 1.08, p=0.008), but this was not the case for participants with a more prosocial SVO angle (+1 SD; 27.64°; B = 1.14, SE = 1.18, p=0.337). This shows that more individualistic individuals made relatively more self-benefitting ('selfish') decisions at increasing levels of perceived stress (*Figure 4*, left panel). Cortisol

(AUC) did not modulate the effect of SVO angle on the proportion of prosocial choices ($B = 0.069$, SE $= 0.093$, p=0.459).

## Perceived stress modulates AI and dACC responses to $SV_{self}$ relative to $SV_{other}$ during effort-based decisions

To identify the neural mechanisms underlying the effects of stress on prosocial behaviour, we analysed brain areas which during the offer period responded to the SV of the chosen option relative to the non-chosen option (i.e. work vs. rest) on each trial. We focused our analysis on the AI and dACC using anatomical masks independently determined by *Lockwood et al., 2022*. Both areas were of strong a priori interest as they have been reliably implicated in effort-based decision-making (*Chong et al., 2017*; *Croxson et al., 2009*; *Engström et al., 2014*; *Lockwood et al., 2022*; *Prévost et al., 2010*) and show a consistent sensitivity to stress (*Ahs et al., 2006*; *Berretz et al., 2021*; *Cerqueira et al., 2007*; *Dedovic et al., 2009*; *Gathmann et al., 2014*; *King et al., 2009*; *Kogler et al., 2015*; *Morgado et al., 2015*; *Pruessner et al., 2008*; *Radley et al., 2005*; *Starcke and Brand, 2012*; *Wang et al., 2005*).

To calculate the SV for each participant on each trial, we used an established model which has consistently shown to best characterise participants' choices on this task (*Lockwood et al., 2017*; *Lockwood et al., 2021*; *Lockwood et al., 2022*). The model contains two separate parabolic effort discounting parameters ($K_{self}$, $K_{other}$) and one temperature parameter ($\beta$):

$$\text{Subjective value} = \text{Reward} - (\text{Discount} \times \text{Effort}^2)$$

$$\text{Discount parameter on self trails} = K_{self}$$

$$\text{Discount parameter on other trails} = K_{other}$$

Maximum likelihood estimation and a model comparison approach confirmed that this was the best-fitting model – in terms of the lowest Bayesian information criterion (BIC; see *Figure 1—figure supplement 1*). We further validated this winning model by performing parameter recovery (see *Supplementary file 1a*). Each participant's $K_{self}$ and $K_{other}$ values were used to calculate the SV of the chosen option relative to the non-chosen option (i.e. work vs. rest) on each trial using the winning model.

These values were then used during the first-level analysis as a parametric modulator to determine which regions scaled with SV for each participant on self trials ($SV_{self}$) and other trials ($SV_{other}$). Using each participant's contrast images for $SV_{self}$ and $SV_{other}$, we extracted the parameter estimates from the regions of interest (ROI) in the AI and dACC, and analysed them using linear mixed-effects models with the *lmer* function in R. The anticipated Group (stress vs. control) by Recipient (self vs. other) interaction was not significant in either ROI (dACC: $B = -0.242$, SE $= 0.486$, p=0.619; AI: $B = -0.719$, SE $= 0.498$, p=0.152). As above, and in light of the heterogeneity of responses to the stress induction, we included perceived stress (AUC) as a between-subject predictor and Recipient as a within-subject predictor. This revealed a significant interaction between perceived stress and Recipient in both the dACC ($B = -0.566$, SE $= 0.237$, p=0.019) and AI ($B = -0.547$, SE $= 0.247$, p=0.029; *Figure 5*). Simple slopes analyses revealed that in the dACC there was a negative association between perceived stress and $SV_{other}$ ($B = -0.413$, SE $= 0.176$, p=0.020) but not between perceived stress and $SV_{self}$ ($B = 0.153$, SE $= 0.176$, p=0.386). Thus, the results suggest that greater perceived stress perturbs responses to $SV_{other}$ in the dACC. Simple slopes analysis in the AI did not reveal any significant associations (all ps>0.096). Cortisol (AUC) did not modulate the effect of Recipient in either ROI (dACC: $B = -0.342$, SE $= 0.250$, p=0.174; AI: $B = -0.225$, SE $= 0.257$, p=0.383).

When linking activation difference in dACC and AI to behaviour, we found that – independent of the stress manipulation – the difference in activation between $SV_{self}$ and $SV_{other}$ in the dACC predicted the proportion of prosocial choices. Thus, greater activation to $SV_{self}$ relative to $SV_{other}$ predicted a lower proportion of prosocial choices ($B = -0.704$, SE $= 0.339$, p=0.041). This relationship was not present in the AI ($B = -0.423$, SE $= 0.332$, p=0.205).

To complement the ROI analyses, we conducted whole-brain analyses with a statistical threshold of p<0.05 family-wise error (FWE) corrected at the cluster level with a cluster defining threshold of p<0.001 across the whole brain. For each participant, we created contrast images for $SV_{self} - SV_{other}$ (1 -1) and $SV_{other} - SV_{self}$ (–1 1). These contrast images were then used in one-sample *t*-tests with Perceived

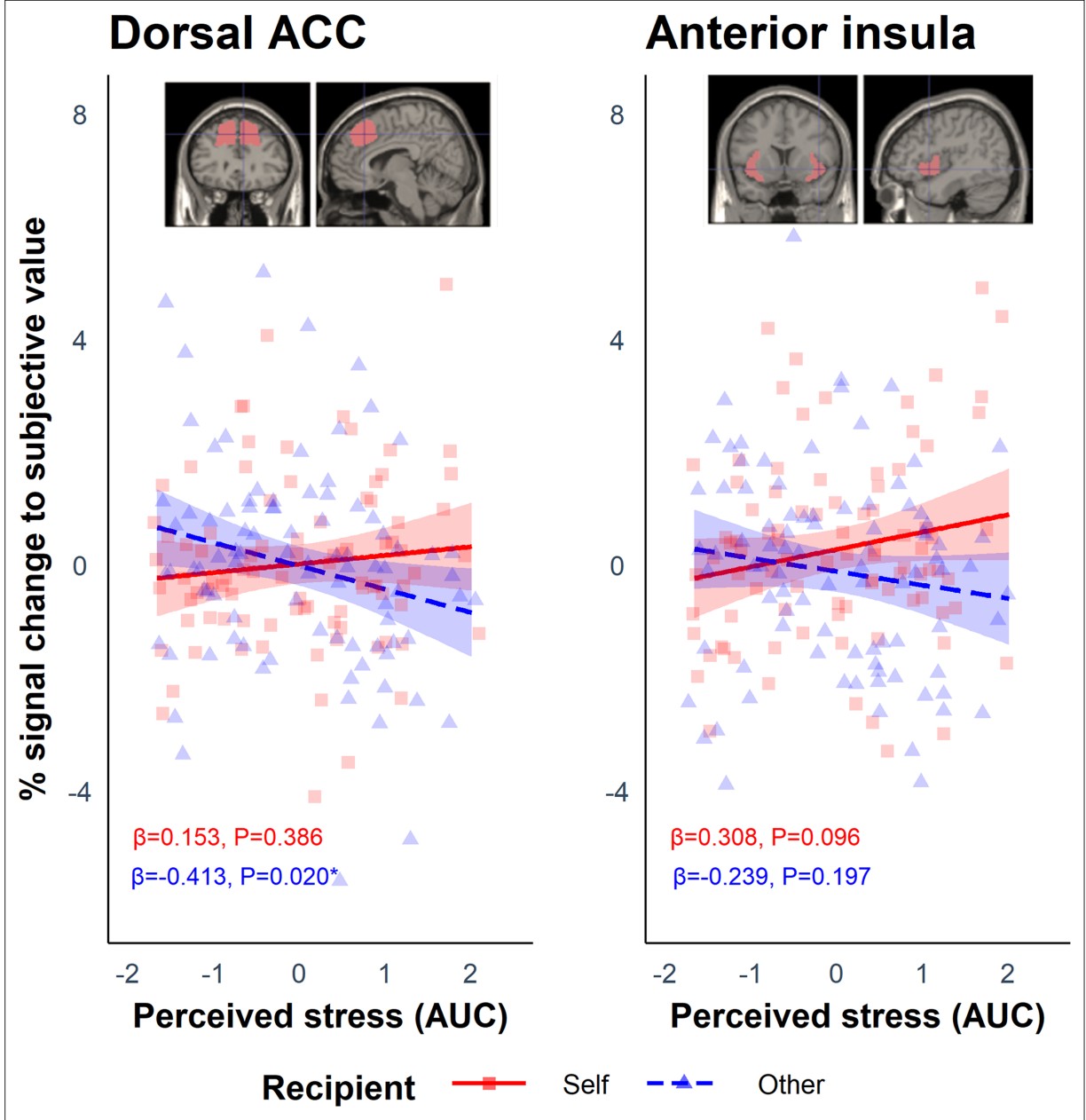

**Figure 5.** Results from the regions of interest (ROI) analysis which showed a significant interaction between Recipient and Perceived stress in the dorsal anterior cingulate cortex (dACC: $B = -0.566$, SE = 0.237, p=0.019) and anterior insula (AI: $B = -0.547$, SE = 0.247, p=0.029). These show the percentage signal change to subjective value (SV) during the offer phase associated with perceived stress on self and other trials. For the dACC, simple slopes analysis revealed a negative association between perceived stress and $SV_{other}$ (blue dashed line) but not between perceived stress and $SV_{self}$ (red solid line). The ribbons represent the 95% confidence intervals. Each point represents the individual data points from the participants for the self (red squares) and other (blue triangles) condition.

Stress as a covariate. This revealed several complementary regions which responded preferentially to $SV_{self}$ compared to $SV_{other}$ at increasing levels of perceived stress, including the dorsolateral prefrontal cortex (dlPFC; middle frontal gyrus: $x = -46$, $y = 20$, $z = 40$; *Table 1*).

As with the behavioural analysis, we investigated whether SVO angle modulated the impact of perceived stress on responses to $SV_{self}$ and $SV_{other}$ in the dACC and AI. We conducted a linear mixed-effects model with SVO angle, Perceived Stress, and Recipient as predictors and the parameter estimates as the dependent variable. As there was no significant three-way interaction between SVO angle, Perceived Stress, and Recipient in either region (ps>0.089), which would have mirrored the

**Table 1.** Regions resulting from a whole-brain analysis in which activity scaled more with SV$_{self}$ compared to SV$_{other}$ that covaried with perceived stress (using a statistical threshold of p<0.05 FWE corrected at the cluster level having thresholded at p<0.001 across the whole brain).

| Brain region | Peak voxel | | | Cluster size (k) | Z |
|---|---|---|---|---|---|
| | x | y | z | | |
| *SV$_{self}$ > SV$_{other}$ covarying with perceived stress* | | | | | |
| R inferior frontal gyrus (pars opercularis) | 44 | 18 | 34 | 430 | 4.61 |
| R midcingulate cortex | 12 | -6 | 44 | 157 | 4.57 |
| L middle occipital gyrus | −38 | −84 | 28 | 135 | 4.23 |
| L thalamus | −10 | −14 | 6 | 124 | 4.14 |
| L middle frontal gyrus | −46 | 20 | 40 | 327 | 3.99 |
| L middle temporal gyrus | −46 | −58 | 14 | 187 | 3.98 |
| R middle occipital gyrus | 44 | −86 | 16 | 144 | 3.91 |

FWE, family-wise error; L, left; R, right; SV, subjective value.

behavioural data, we ran simpler models including only the interactions between SVO angle and Recipient as well as SVO angle and Perceived Stress. In the dACC, there was a significant interaction between Perceived Stress and SVO angle ($B = −0.044$, SE = 0.018, p=0.017) (*Figure 4*, middle panel). Simple slopes analyses revealed that those with a more prosocial SVO (+1 SD; 27.35°) showed a significant decline in activation to SV at increasing levels of perceived stress ($B = −0.537$, SE = 0.200, p=0.009), but this was not the case for participants with a more individualistic SVO (–1 SD; 11.00°; $B = 0.182$, SE = 0.189, p=0.337). There was also a significant interaction between SVO angle and Recipient in the dACC ($B = 0.061$, SE = 0.028, p=0.034) (*Figure 4*, right panel). Simple slopes analyses revealed a positive association between SVO angle and SV$_{other}$ ($B = 0.043$, SE = 0.021, p=0.041) but not SV$_{self}$ ($B = −0.018$, SE = 0.021, p=0.399). Thus, an increase in dACC activation to SV$_{other}$ relative to SV$_{self}$ was associated with a more prosocial SVO angle. In the AI, there was no significant interactions involving SVO angle (ps>0.126).

## Discussion

Much of our prosocial behaviour entails a willingness to exert effort. We investigated how acute stress – a prevalent everyday occurrence – impacts effortful prosocial behaviour and report three important findings. Firstly, prosocial behaviour under stress was dependent on the amount of effort involved. Compared to participants in the control group, those in the stress group were less willing to exert effort for other-benefitting rewards at a relatively low level of effort. Secondly, by using neuro-imaging combined with computational modelling, we found that participants who perceived more stress during the experiment showed increased activation in dACC and AI to SV on self- relative to other-benefitting trials. Thirdly, the detrimental effects of acute stress on effortful prosocial behaviour were especially marked in more selfish participants, who also showed differences in dACC activation. This reveals how individual differences in prosocial tendencies can shape prosocial behaviour under acute stress.

Our task separated the effect of acute stress on participants' willingness to exert effort for self-benefitting, compared to other-benefitting, rewards (*Lockwood et al., 2017*; *Lockwood et al., 2021*; *Lockwood et al., 2022*). Thus, changes in prosocial behaviour were not only the result of general changes in motivation to exert effort following stress (*Bogdanov et al., 2021*; *Bryce and Floresco, 2016*; *Shafiei et al., 2012*). Moreover, in previous studies using economic games in which participants could make financial transfers to others (for a meta-analysis, see *Nitschke et al., 2022a*), it was not clear whether differences in prosocial behaviour were the result of changes in reward sensitivity (e.g. how participants value their own monetary gains following stress; *Berghorst et al., 2013*; for a meta-analysis, see *Forbes et al., 2023*), or the result of actual changes in prosocial motivation. As we

manipulated effort costs, rather than financial costs, we showed that acute stress specifically changed participants' *motivation* to help others when the effort costs involved in gaining rewards for others were relatively low (i.e. effort level 2; 40% of participant's MVC). Increased or unchanged prosocial behaviour under stress could therefore be restricted to situations in which effort costs are absent or when participants are helping close others (*Margittai et al., 2015*; *Margittai et al., 2018*).

A potential explanation for why we found group differences in other-benefitting relative to self-benefitting effortful behaviour at a lower level of effort and not at higher levels is that acute stress also had a general effect on participants' willingness to exert effort as shown by the presence of a Group × Effort interaction (see also *Figure 3*). At higher effort levels, any effects of recipient (i.e. self vs. other) may have been masked by a more general effect of acute stress on participants' willingness to exert effort. Thus, future studies investigating differences in prosocial motivation under stress will need to separate general reductions in motivation from specific differences in prosocial motivation. One way to achieve this could be to investigate lower effort levels in a more fine-grained manner.

We used a computational modelling approach combined with neuroimaging to provide insight into potential neural mechanisms underlying the behavioural results. At increasing levels of perceived stress, the dACC and AI showed greater activation to SV on self-benefitting ($SV_{self}$) relative to other-benefitting trials ($SV_{other}$). In the dACC, this difference was driven by reduced activation to $SV_{other}$ at increasing levels of perceived stress. Both the dACC and AI have been consistently implicated in effort-based decision-making (*Chong et al., 2017*; *Croxson et al., 2009*; *Engström et al., 2014*; *Lockwood et al., 2022*; *Prévost et al., 2010*; *Rudebeck et al., 2006*; *Walton et al., 2003*) and are sensitive to stress effects (*Berretz et al., 2021*; *Gathmann et al., 2014*; *Kogler et al., 2015*; *Morgado et al., 2015*; *Starcke and Brand, 2012*). The dACC is thought to track effort costs in a domain-general manner - regardless of whether effort costs are physical or cognitive (*Chong et al., 2017*), or whether effort is being exerted for one's own or someone else's benefit (*Contreras-Huerta et al., 2020*; *Lockwood et al., 2022*). Thus, reduced tracking of SV in the dACC on other-benefitting trials under heightened stress could reflect a reduced priority placed on other people's gains.

It is important to acknowledge that we did not find the anticipated Group by Recipient interaction in either the AI or dACC; the effect of Recipient on parameter estimates was only modulated by perceived stress (AUC) – the amount of stress participants reported across the experiment. Similarly, we did not see the anticipated Group by SVO angle interaction for the proportion of prosocial choices, but only an interaction between perceived stress and SVO angle. This absence of a group effect may have resulted from several participants in the stress group not responding to the stressor and, conversely, those in the control group being stressed, for example, by the scanning environment itself (*Noack et al., 2019*). Heterogeneity in stress responses is a feature of all studies involving acute stress as no stress induction protocol is completely effective, particularly within a scanning environment where successful stress induction (as measured by cortisol reactivity) ranges from 47.5 to 65.0% for the MIST (*Noack et al., 2019*).

One potential mechanism underlying a shift in prosocial tendencies under greater perceived stress could be a downregulation of the brain's 'executive control network' (*Hermans et al., 2014*). The dlPFC is a key component of this network, and, in our complementary whole-brain analysis, this area showed reduced activation to $SV_{other}$ relative to $SV_{self}$ at increasing levels of perceived stress. Similar changes in dlPFC activation following stress have been reported during working memory tasks (*Qin et al., 2009*). Three meta-analyses (*Bellucci et al., 2020*; *Cutler and Campbell-Meiklejohn, 2019*; *Rhoads et al., 2021*) have highlighted the dlPFC as a key region involved in prosocial behaviour (for an overview, see *Lamm and Forbes, 2023*) potentially due to its role in inhibiting selfish responses (*Feng et al., 2015*; *Strang et al., 2015*). The dlPFC has also been implicated in cognitive flexibility under acute stress. For example, *Kalia et al., 2018* used functional near-infrared spectroscopy to show that reduced cognitive flexibility under stress was related to changes in activation in the dlPFC in men. In our study, participants in the control group were more likely to exert effort for self rewards compared to other rewards at higher, but not at lower, levels of effort. Whilst participants in the stress group favoured exerting effort for self rewards at every effort level (*Figure 3*). This consistent preference for self rewards compared to other rewards at all effort level suggests that stressed participants did not adapt their social behaviour in response to changing contextual information. This supports multiple studies showing reduced cognitive flexibility under stress (*Goldfarb et al., 2017*; *Kalia et al., 2018*; *Raio et al., 2017*; *Shields et al., 2016*). An exciting avenue for future work is to

test whether individual differences in executive functions, such as inhibition and cognitive flexibility, predict changes in social behaviour following acute stress. This would be analogous to the finding in non-social domains, where greater working memory capacity protects against stress-induced changes in learning (*Otto et al., 2013*).

Individual differences in prosocial traits as measured by SVO (*Murphy et al., 2011*) modulated the effect of perceived stress on effortful prosocial behaviour. Participants with a more individualistic SVO chose to exert effort more often for rewards for themselves compared to rewards for others at increasing levels of perceived stress. This relationship was not seen in individuals with a more prosocial SVO. Similar results have been reported when participants shared money with others under time constraints, whereby time pressure exaggerates existing prosocial tendencies (*Chen and Krajbich, 2018*; but see *Bouwmeester et al., 2017*). Our findings lend further support to models of acute stress that emphasise a shift to automatic or habitual responding (*Hermans et al., 2014*; *Starcke and Brand, 2012*) and suggest that such general shifts in cognitive processing also apply within the social domain. The specific consequences these shifts to more automatic processing have for social behaviour are likely to be person- and context-specific, and understanding these factors should be a key aim for future work (*Nitschke et al., 2022a*).

Our study extends previous findings demonstrating the importance of individual differences in empathy (*Azulay et al., 2022*), mentalising (*Schulreich et al., 2022*), and SVO (*Ying et al., 2022*) in predicting prosocial responses under acute stress. However, it is not yet clear when acute stress accentuates or suppresses existing prosocial tendencies. For example, *Schulreich et al., 2022* found that increases in cortisol were associated with *reductions* in donations in participants scoring high on mentalising, whereas *Azulay et al., 2022* found that increases in cortisol predicted *enhanced* donations in participants with high-trait empathy (see also *Speer et al., 2022*). Given the heterogeneity in behaviour in economic games under stress (*Nitschke et al., 2022a*), future studies may benefit from manipulating the effort involved in helping, or other non-financial costs, to help us understand how the social behaviour of different individuals is affected by acute stress.

The neuroimaging data showed – independent of our stress manipulation – that SVO angle predicted increased responses to $SV_{other}$ but not to $SV_{self}$ (*Figure 4*, right panel). This is in line with previous work showing that SVO modulates dmPFC/dACC activity depending on whether decisions are social or self-interested (*Kuss et al., 2015*). Additionally, participants with a more prosocial SVO showed reduced responses in the dACC to SV (across both self and other trials) at greater levels of perceived stress (*Figure 4*, middle panel). This suggests that more prosocial individuals may become less sensitive to SV overall following stress, whilst the responses of more individualistic participants to SV do not change under stress. Trying to link these activation differences to changes in effortful prosocial behaviour is difficult given the absence of the three-way interaction between SVO angle, Perceived Stress, and Recipient, which would have mirrored the behavioural results. Overall, differences in activation between $SV_{self}$ and $SV_{other}$ in the dACC predicted the proportion of prosocial choices, so greater activation to $SV_{self}$ relative to $SV_{other}$ predicted a lower proportion of prosocial choices. Thus, it remains unclear how activation differences to SV across both self trials and other trials relates to changes in prosocial behaviour under stress. *Schulreich et al., 2022* found that a decline in charitable donations following increases in cortisol in high mentalisers was related to a reduced representation of value for donations in the right dlPFC. Whilst there are important differences between the present study and *Schulreich et al., 2022*, such as the way in which prosocial behaviour was measured, both studies suggest that existing differences in social preferences and abilities (i.e. mentalising, SVO) can have a detrimental effect on the neural representations of value following acute stress. Establishing how these changes in neural representations of value impact behaviour following acute stress is a challenge for future work.

## Future directions and implications

Participants earned money for either themselves or the next participant in the study who was not known to them. Future work should manipulate the social distance of the other recipient. This will determine whether participants under stress are more willing to put in effort for close others, such as family or friends (*Margittai et al., 2015*; *Margittai et al., 2018*). Additionally, it will be important to explore whether we see similar reductions in prosocial motivation under stress when preventing harm in others (*Hartmann et al., 2022*) rather than gaining rewards for them.

Since there are known sex/gender differences in social behaviours following acute stress (*Nitschke et al., 2022b*; *Tomova et al., 2014*; *Zhang et al., 2019*), we only tested young men in the current study (see 'Materials and methods'). Age has also been shown to modulate effortful prosocial behaviour (*Cutler et al., 2021*; *Lockwood et al., 2021*). Thus, future studies should investigate how effortful prosocial behaviour in women and in older populations is affected by acute stress. Our power calculation was based on a 2 × 2 design (Group × Recipient); however, several of our key findings involved three-way interactions (e.g. between Group, Recipient, and Effort). Thus, future studies should aim to replicate our effects with larger sample sizes to ensure the robustness of these effects.

The experience of stress is a common everyday occurrence (*Hassard et al., 2018*), and the negative effects of prolonged stress for our physical and mental health are well-documented (*Lecrubier, 2001*). Our results show that acute stress is not only potentially detrimental for our own well-being but can have negative consequences for our behaviour towards others. Given the importance of effortful prosocial behaviour for relationships and, more generally, for social cohesion, this highlights the importance of tackling the major sources of stress, such as low income (*Haushofer and Fehr, 2014*) and inequality (*Pickett and Wilkinson, 2010*), as well as providing individuals with the resources to cope with stress when it does occur (e.g. *Meichenbaum, 2017*).

## Conclusion

Our study demonstrates that participants under acute stress were less willing to exert effort for another person's benefit at a low level of effort. The adverse effects of acute stress on prosocial effortful behaviour were most marked in more selfish individuals. These findings show that using effort-based tasks and incorporating trait differences in prosocial tendencies could be key to understanding changes in prosocial behaviour under acute stress. Additionally, we found that the effects of stress were related to activation differences in the AI and dACC – areas strongly implicated in effort-based decisions and sensitive to stress effects. Thus, we provide valuable insight into the potential neural mechanisms underlying stress-induced changes in effortful prosocial behaviour. Moreover, the results raise the possibility that shifts to automatic or habitual responding under stress could also apply within the social domain (*Hermans et al., 2014*). Together, this emphasises the need to tackle the sources of stress and provide individuals with the resources to cope with it, so that the potential detrimental effects of stress on prosocial behaviour are curtailed.

## Materials and methods

### Participants

We recruited healthy, right-handed, male students who were non-smokers or smoked less than five cigarettes per day. Only male participants were included as previous work has shown that they show a stronger salivary glucocorticoid response to laboratory stressors (*Kirschbaum et al., 1999*). We did not recruit participants who had studied or were studying psychology nor those who had taken part in previous stress induction studies. To achieve 80% power to detect a small-to-medium effect size ($f = 0.15$; $\eta^2 = 0.022$) in a between-subject (Group: stress vs. control) by within-subject (Recipient: self vs. other) design (calculated using G*Power, *Faul et al., 2007*), we required a sample size of at least 90 participants. To account for potential exclusions, we recruited 96 participants in total.

Three participants were excluded due to either a disruption during the scanning (n = 1), having previously worked in the lab (n = 1), or never choosing to work on other trials (n = 1). This left a sample of 93 participants (mean age = 23.5 years, SD = 2.89, range = 18–31) for the behavioural analysis with 47 in the stress group and 46 in the control group. The study was approved by the ethics committee of the University of Vienna (reference number: 00412), and all participants provided written informed consent. Participants were paid 40 euros for their participation plus the money they received from the prosocial effort task (between 6 and 8 euros).

### Procedure

Participants came to the lab twice. In the first session, participants were familiarised with the scanning environment and completed a series of online questionnaires, including SVO (*Murphy et al., 2011*; *Murphy and Ackermann, 2014*). Participants were told not to consume any alcohol, cigarettes, or medication, nor to engage in vigorous exercise 24 hr before the second session, and, 2 hr before the

session, not to consume any food or drinks other than water. During the second session, participants completed the stress induction procedure (or a control procedure) and prosocial effort task within the MRI scanner. Saliva samples and visual analogue scales were used throughout the second session to measure cortisol and perceived stress. The second session was within approximately 1 wk of the first session, lasted 160 min, and always took place between 12:00 and 18:00 to account for diurnal fluctuations in cortisol.

## Stress induction

Participants in the stress group completed an adapted version of the Montreal Imaging Stress Test (MIST; *Dedovic et al., 2005*; *Tomova et al., 2017*) in which they completed challenging mental arithmetic questions under time pressure. During the MIST, participants saw their live and, seemingly, 'below average' performance on a scale at the top of the screen and a video stream of the experimenters observing their performance. This ensured social evaluation from the experimenters who also reminded participants that they could not use their data if their performance did not reach the group average. Throughout the scanning session, participants completed three 6-min blocks of the MIST and also completed a counting task three times in 2-min blocks (*Kirschbaum et al., 1993*). Here, participants were required to count backwards in steps of 13 or 17 from a large number (e.g. 2053, 2036, 2019, etc.). The experimenters asked participants to start the count again if they made a mistake or speed up if responses were not quick enough. Participants in the control group completed the same mental arithmetic questions in the MIST but without the time pressure and social evaluation. The experimenters wore white lab coats in the stress condition but not in the control condition. For the counting task, the control group counted silently in steps of either 5, 10, or 20 for 2 min and did not receive feedback from the experimenters. Participants completed a MIST block, a run of the prosocial effort task, the counting task, and another run of the prosocial effort task, and this sequence was repeated three times during the scanning period. This ensured that participants' psychological and physiological stress responses were maintained throughout the scanning period (*Figure 2*).

## Cortisol and stress measurements

To measure cortisol responses, saliva samples were collected throughout the experiment using oral swabs (Sarstedt Salivette) which were placed in participants' mouths for 2 min. Six saliva samples were collected (all times are relative to the start of stress induction): upon arrival (sample 1; –42 min), before the first block (sample 2; 0 min) of the MIST after the second MIST (sample 3; 38 min) and during the third MIST (sample 4; 64 min), at the end of the scan (sample 5; 94 min), and then following debrief (sample 6; 119 min). Following each session, saliva samples were frozen and stored at –20°C until analysis. During each saliva sample, we asked participants to complete a visual analogue scale to determine their perceived stress levels ('Right now I feel stressed') on a 7-point scale from 'not at all' (1) to 'very much' (7). Additionally, participants completed these ratings following the first MIST (rating 3; 11 min) and following the second counting task (rating 5; 51 min), although saliva samples were not collected at these timepoints (see *Figure 2*).

After thawing, the oral swabs were centrifuged at 3000 rpm for 5 min, which resulted in a clear supernatant of low viscosity. Salivary concentrations were measured using commercially available chemiluminescence immunoassay with high sensitivity (IBL International, Hamburg, Germany). The intra- and inter-assay coefficients for cortisol were both below 9%. Three subjects were excluded from the cortisol analysis as they failed to provide sufficient saliva in five, three, and two (out of a possible six) saliva samples, respectively. One participant did not provide sufficient saliva for his fifth sample, but this value was interpolated using his fourth and sixth samples to allow AUC to be calculated. This created a final sample of 91 participants (stress = 46, control = 45) whose saliva samples were analysed. Three participants had a missing visual analogue scale stress rating, and these were interpolated in the same manner to allow AUC to be calculated.

## Prosocial effort task

We used the prosocial effort task developed by *Lockwood et al., 2017*. On each trial, participants had a choice between two options: *work* or *rest*. The *rest* option always involved receiving a low reward for no effort (one credit for no effort), whereas the *work* option involved putting in physical effort for a higher reward (more credits for more effort). The *rest* option was always the same, whereas

the effort level and reward level for the *work* option differed on each trial. There were five reward levels – 2, 4, 6, 8, or 10 credits – and these credits were converted to money at the end of the study. There were five effort levels – 30, 40, 50, 60, or 70% of participants' MVC – and the effort level was indicated by how filled the circle was, that is, the more filled the circle, the more effort was required. Each participants' MVC was collected at the start of the study by asking them to squeeze an MR-compatible hand dynamometer (Current Designs Inc, Philadelphia, USA) as hard as possible. Thus, the effort level was calibrated to each participants' individual strength. Before and immediately after the experiment, participants experienced each effort level and rated (1) how much effort they exerted, (2) how physically demanding it was, and (3) how uncomfortable it was to squeeze to the required effort level on a 21-point Likert scale.

Each trial started with a variable jitter (2–3.5 s), after which participants were presented with the binary choice between *work* and *rest*. They had 3.2 s to make their decision by using their left hand to press the button box. After 3.2 s, their choice was highlighted with a yellow box for 0.3 s. If they chose to rest, then they did not do anything for the remainder of the trial and received one credit at the end of it. If they chose to work, then following a variable jitter (2–3.5 s), participants had to squeeze the hand dynamometer at the required effort level for 1 s during a 3-s window. During the squeezing period, a bar was displayed on the screen which showed the amount of force participants were currently exerting and a yellow line indicated the required effort level. If participants squeezed above the yellow line for at least 1 s, then following another variable jitter (1–2 s), the credits they received (or the next participants received) were displayed on the screen (0.5 s). If participants failed to squeeze above the yellow line during the 3-s period, then they received no credits. If participants failed to respond within 3.2 s during the offer period, then '0 credits – Please respond quicker!' was displayed on the screen for the duration of the trial.

On half the trials, participants made decisions whether to put in effort to earn more money for themselves (*self* trials), and on the other half of trials, they could put in effort to earn money for the next participant in the study (*other* trials). On *self* trials, 'You' (German: '*Du*') appeared in the middle of the screen and the stimuli were red, whereas, on *other* trials 'Thomas' (i.e. the name of the next participant) was displayed in the middle of the screen and the stimuli were presented in blue (**Figure 1**). At the end of the study, participants received the money they earned for themselves plus the money the previous participant had earned for them. Each combination of effort level and reward level was presented three times in *self* trials and three times in *other* trials. This created a total of 150 trials presented in a pseudorandomised order and split into six runs of 25 trials. The task was presented using the Cogent toolbox (http://www.vislab.ucl.ac.uk/cogent.php) in MATLAB (MathWorks).

## Computational modelling of choices

We used a model comparison approach (*Lockwood et al., 2017*) in which we compared models with a single discount parameter, $K$, to those with a separate discount parameter for self ($K_{self}$) and other ($K_{other}$), and models with a single noise parameter, $\beta$, to those with a separate noise parameter for self ($\beta_{self}$) and other ($\beta_{other}$) trials. We also compared models in which rewards were discounted linearly (subjective value = reward - effort * k), hyperbolically (subjective value = reward/[1 + [effort * k]]), and parabolically (subjective value = reward – effort$^2$ * k). In total, these combinations (2 × 2 × 3) created 12 different models which we compared using the BIC based on the log-likelihood. The discount parameter, $k$, was bounded from 0 to 1.5 to ensure an appropriate range.

## FMRI acquisition and analysis

MRI data were collected using a 3 Tesla MRI scanner (Skyra, Siemens Medical) and a 32-channel-head coil. The structural scans were acquired using a magnetization prepared rapid gradient echo (MPRAGE) sequence with the following parameters: TR = 2300 ms; TE = 2.43 ms; flip angle = 8°, voxel size = 0.8 mm isotropic; field of view (FOV) = 240 × 240 mm. BOLD functional scans were acquired with a multiband accelerated EPI sequence with the following parameters: TR = 1200 ms; TE = 34 ms; flip angle = 66°; slices = 52; multiband acceleration factor = 4 (i.e. 13 excitations per TR); FOV = 210 × 210 mm, voxel size = 2 × 2 × 2 mm.

## Preprocessing

Data were preprocessed and analysed using SPM12 in MATLAB (http://www.fil.ion.ucl.ac.uk/spm/). The functional images were slice-time corrected to the middle slice, realigned to the mean image, and smoothed with a Gaussian kernel (5 mm full-width at half maximum). The structural scan was co-registered to the mean functional scan from the first run and then segmented into grey matter, white matter, cerebrospinal fluid, bone, soft tissue, and air. Diffeomorphic Anatomical Registration Through Exponentiated Lie Algebra (*Ashburner, 2007*) was used to normalise both the structural and functional scans to the Montreal Neurological Institute template.

Next, we looked at head motion using framewise displacement (FD; *Power et al., 2012*) and excluded four participants who showed FD > 0.5 mm in over 35% of scans in two or more runs. Two particpiants showed an FD > 0.5 mm in over 35% of scans in run 6. These runs were excluded but the two participants remained in the analysis. Data from run 2 was missing for one participant due to a technical issue during the scan, so this run was also not included in the neuroimaging analysis. Finally, as outlined above, one subject in the stress group never chose to exert effort for the other, so was excluded from the analysis. This left a final sample of 89 participants (45 control; 44 stress) for the imaging analysis, with 3 out of these 89 participants missing one run from the six.

## FMRI design

To create our design matrix, regressors were constructed for three events in the trial – the offer phase, force phase, and outcome phase (*Lockwood et al., 2022*) – and these were convolved with SPM's canonical haemodynamic response function. Each regressor had an associated parametric modulator: for the offer phase, this was the SV of the chosen option relative to SV of the non-chosen option; for the force phase, this was the chosen effort level (0:5); and for the outcome phase, this was the reward level (i.e. the number of credits) received at the end of the trial. Our hypotheses and research questions concerning the effects of acute stress concerned the offer phase, that is, when participants were deciding whether to exert effort or not (work vs. rest). Therefore, we limited our reporting to this event. We split the trials according to the recipient of the reward, so there were separate parametric regressors for self and other trials. If a participant did not respond quickly enough when making their choice (within 3.2 s), then this trial was labelled as a missed trial and was included as an additional regressor.

Six head motion parameters were included in the general linear model (GLM) as were scrubbing regressors (*Power et al., 2014*). Functional scans were individually 'scrubbed' if they showed an FD > 0.5 mm and these were included as nuisance regressors in the first-level analysis. Data were high-pass filtered with a 128 s cutoff as is standard in SPM, and all six runs were combined into one GLM.

For each of the parametric modulators, we created first-level images for the self > other (1 -1) and other > self (–1 1) contrast. This allowed us to look at regions which responded more on self trials compared to other trials and vice versa. These images were then inputted into our second-level analyses in which we conducted one-sample *t*-tests with perceived stress (AUC) as a covariate. This whole-brain analysis used a statistical threshold of p<0.05 FWE corrected at the cluster-level having thresholded at p<0.001 across the whole brain. We used the REX toolbox (http://web.mit.edu/swg/software.htm) to extract parameter estimates from two ROIs, the dACC and AI, which were taken from *Lockwood et al., 2022*.

## Acknowledgements

We thank Livia Tomova for guidance in designing the stress induction protocol and Jonas Nitschke for providing extensive feedback on a previous version of the manuscript.

## Additional information

### Funding

| Funder | Grant reference number | Author |
| --- | --- | --- |
| Austrian Science Fund | I3381 | Claus Lamm |

| Funder | Grant reference number | Author |
|---|---|---|
| Austrian Science Fund | P34775-B | Isabella C Wagner |
| Biotechnology and Biological Sciences Research Council | BB/R010668/1 | Matthew AJ Apps |
| Medical Research Council | MR/ P014097/1 | Patricia L Lockwood |
| Medical Research Council | MR/P014097/2 | Patricia L Lockwood |
| Wellcome Trust | Sir Henry Dale Fellowship | Patricia L Lockwood |
| Jacobs Foundation | Research Fellowship | Patricia L Lockwood |
| Royal Society | 223264/Z/21/Z | Patricia L Lockwood |

The funders had no role in study design, data collection and interpretation, or the decision to submit the work for publication. For the purpose of Open Access, the authors have applied a CC BY public copyright license to any Author Accepted Manuscript version arising from this submission.

## Author contributions

Paul AG Forbes, Conceptualization, Data curation, Formal analysis, Validation, Investigation, Visualization, Methodology, Writing – original draft, Project administration, Writing – review and editing; Gökhan Aydogan, Conceptualization, Methodology, Writing – review and editing; Julia Braunstein, Boryana Todorova, Investigation, Methodology, Project administration, Writing – review and editing; Isabella C Wagner, Formal analysis, Methodology, Writing – review and editing; Patricia L Lockwood, Matthew AJ Apps, Conceptualization, Formal analysis, Visualization, Methodology, Writing – review and editing; Christian C Ruff, Conceptualization, Resources, Supervision, Funding acquisition, Methodology, Project administration, Writing – review and editing; Claus Lamm, Conceptualization, Resources, Supervision, Funding acquisition, Investigation, Methodology, Project administration, Writing – review and editing

## Author ORCIDs

Paul AG Forbes ⓘ https://orcid.org/0000-0002-0138-8508
Julia Braunstein ⓘ http://orcid.org/0000-0002-0006-6590
Boryana Todorova ⓘ http://orcid.org/0000-0003-4840-498X
Isabella C Wagner ⓘ https://orcid.org/0000-0002-4383-8204
Christian C Ruff ⓘ https://orcid.org/0000-0002-3964-2364
Claus Lamm ⓘ http://orcid.org/0000-0002-5422-0653

## Ethics

This research was approved by the ethics committee of the University of Vienna (reference number: 00412). All participants gave informed consent before taking part in the study.

Joint Public Review: https://doi.org/10.7554/eLife.87271.3.sa1
Author Response https://doi.org/10.7554/eLife.10.7554/eLife.87271.3.3.sa2

# Additional files

## Supplementary files

• Supplementary file 1. Parameter recovery, effort rating correlations, and full output of the mixed models. (a) Parameter recovery. We stimulated data based on our trial structure and determined whether the parameters were recoverable for the best-fitting model. There was excellent recovery as showed by the strong relationship between fitted and simulated (or 'true') parameters. (b) Spearman's rho correlations between the difference in how physically demanding participants rated the effort levels after the experiment relative to before and the number of effortful choices in each group for each recipient. p-Values are shown in brackets. There were no significant correlations. (c) Type III Wald test on choice data from the GLMM. The binary dependent variable was choice (0 = rest, 1 = work). Group, Recipient, Effort, Reward, and their interactions were fixed effects. We included a subject-level random intercept and a random slope for Recipient. Significant results are

shown in bold. (d) Post hoc comparisons of choice data. The interaction between Group and Effort, and the interaction between Group, Recipient, and Effort, is shown. All p-values are Bonferroni corrected, and significant results are shown in bold. Means were extracted using the emmeans package in R (*Lenth, 2022*). (e). Type III Wald test on force data from the LMM. Group, Recipient, Effort, Reward, and their interactions were fixed effects, and force was the dependent variable – area under the curve during the force period relative to each participants' MVC. We included a subject-level random intercept and a random slope for Recipient. Significant results are shown in bold.

- MDAR checklist

## Data availability

Data and analysis code are available on the Open Science Framework https://osf.io/4bqf7/.

The following dataset was generated:

| Author(s) | Year | Dataset title | Dataset URL | Database and Identifier |
|-----------|------|---------------|-------------|-------------------------|
| Forbes P | 2023 | Acute stress and prosocial effort | https://osf.io/4bqf7/ | Open Science Framework, 4bqf7 |

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
