## [Editor Report · eLife assessment]

This study reports **useful** findings on the influence of acute stress on prosocial behavior and its neural correlates. The approach is **solid**, combining neuroimaging and neuroendocrine measures with computational cognitive modeling. The results will be of interest to researchers seeking to better characterize the influence of stress on neural computations mediating complex social behavior.

---

## [Referee Report · Joint Public Review]

This study sought to characterize the influence of acute stress on prosocial behavior, combining an effort-based task with neuroimaging, neuroendocrinological measures, and computational cognitive modeling. Two major results are reported: (1) Compared to controls, participants who experienced acute stress were less willing to exert effort for others, with more prominent effects for those who were more selfish; (2) More stressed participants exhibited an increase in activation in the dorsal anterior cingulate cortex and anterior insula, which are implicated in self-benefiting behavior. The approach is sophisticated and the findings are informative. Concerns regarding potential confounds and data reporting were addressed in a revised submission.

---

## [Author Response]

The following is the authors’ response to the original reviews.

**Public Reviews:**

**Reviewer #1 (Public Review):**
The authors sought to understand the neurocomputational mechanisms of how acute stress impacts human effortful prosocial behavior. Functional neuroimaging during an effort-based decision task and computational modeling were employed. Two major results are reported: 1) Compared to controls, participants who experienced acute stress were less willing to exert effort for others, with a more prominent effect for those who were more selfish; 2) More stressed participants exhibited an increase in activation in the dorsal anterior cingulate cortex and anterior insula that are critical for self-benefiting behaviour. The authors conclude that their findings have important insights into how acute stress affects prosociality and its associated neural mechanisms.Overall, there are several strengths in this well-written manuscript. The experimental design along with acute stress induction procedures were well controlled, the data analyses were reasonable and informative, and the results from the computational modeling provide important insights (e.g., subjective values). Despite these strengths, there were some weaknesses regarding potential confounding factors in both the experimental design and methodological approach, including selective reporting of only some aspects of this complex dataset, and the interpretation of the observations. These detract from from the overall impact of the manuscript. In particular, the stress manipulation and pro-social task are both effortful, raising the possibility that stressed participants were more fatigued. Other concerns include the opportunity for social dynamics or cues during task administration, the baseline social value orientation (SVO) in each group, and the possibility of a different SVO in individuals with selfish tendencies. Finally, Figure 4 should specify whether the depicted prosocial choices include all five levels of effort.

We thank the reviewer for their comments and suggestions. In our response to the recommendations for the author below, we have dealt with the reviewer’s concerns:

we added additional analysis on the role of fatigue and block effects to the supplementary materials.we provided further information about the role of social cues and dynamics during task administration.we showed there were no baseline group differences in SVO angle.we clarified that Figure 4 refers to the proportion of prosocial choices across all effort levels.

**Reviewer #2 (Public Review):**
This manuscript describes an interesting study assessing the impact of acute stress on neural activity and helping behavior in young, healthy men. Strengths of the study include a combination of neuroimaging and psychoneuroendocrine measures, as well as computational modeling of prosocial behavior. Weaknesses include complex, difficult to understand 3-way interactions that the sample size may not be large enough to reliably test. Nonetheless, the study and results provide useful information for researchers seeking to better understand the influence of stress on the neural bases of complex behavior.The stressor was effective at eliciting physiological and psychological stress responses as shown in Figure 2.Higher perceived stress in more selfish participants (lower social value orientation (SVO) angle) was associated with lower prosocial responding (Figure 4). How can we reconcile this finding with the finding (presented on page 15) that those with a more prosocial SVO showed a significant decline in dACC activation to subjective value at increasing levels of perceived stress? This seems contrary to the behavioral response.A larger issue with the study is that the power analysis presented on page 23 is based on a 2 (between: stress v. control) by 2 (within: self v. other) design. Most of the reported findings come from analyses of 3-way interactions. How can the readers have confidence in the reliability of results from 3-way interaction analyses, which were not powered to detect such effects?

We thank the reviewer for their comments and suggestions. When considering the influence of dACC activation on the behavioural response (i.e., proportion of prosocial choices), it is important to consider the difference in activation to SVself relative to SVother:

The difference in activation to SVself relative to SVother negatively predicted the proportion of prosocial choices, so more activation to SVself relative to SVother predicted a lower proportion of prosocial choices.Similarly, SVO angle negatively predicted the difference in activation to SVself relative to SVother, so more activation to SVself relative to SVother was related to a lower (more individualistic) SVO angle (this is shown by the interaction between Recipient and SVO angle in Figure 4; right panel).In both cases, differences in prosociality (i.e. SVO angle or the proportion of prosocial choices) were related to differences in dACC activation to SVself relative to SVother.

Thus, we agree the finding that those participants with a more prosocial SVO showed a significant decline in dACC activation to SV overall (across SVself and SVother) at increasing levels of perceived stress is difficult to interpret. We expected a three-way interaction between Recipient, SVO angle and Perceived Stress to mirror the behavioural results, rather than a two-way interaction between SVO angle and Perceived Stress. We have now acknowledged this in the Discussion, whilst also highlighting the work of Schulreich et al. (2022) who report a related finding.

We have now added the following section to the results:

“When linking activation difference in dACC and AI to behaviour, we found that – independent of the stress manipulation – the difference in activation between SVself and SVother in the dACC predicted the proportion of prosocial choices. Thus, greater activation to SVself relative to SVother predicted a lower proportion of prosocial choices (B=-0.704, SE=0.339, P=0.041). This relationship was not present in the AI (B=-0.423, SE=0.332, P=0.205).”

And we have added the following to the discussion:

“Additionally, participants with a more prosocial SVO showed reduced responses in the dACC to SV (across both self and other trials) at greater levels of perceived stress (Figure 4; middle panel). This suggests that more prosocial individuals may become less sensitive to SV overall following stress, whilst the responses of more individualistic participants to SV do not change under stress. Trying to link these activation differences to changes in effortful prosocial behaviour is difficult given the absence of the three-way interaction between SVO angle, Perceived Stress and Recipient, which would have mirrored the behavioural results. Overall, differences in activation between SVself and SVother in the dACC predicted the proportion of prosocial choices, so greater activation to SVself relative to SVother predicted a lower proportion of prosocial choices. Thus, it remains unclear how activation differences to SV across both self trials and other trials relates to changes in prosocial behaviour under stress. Schulreich et al. (2022) found that a decline in charitable donations following increases in cortisol in high mentalisers was related to a reduced representation of value for donations in the right dlPFC. Whilst there are important differences between the present study and Schulriech et al. (2022), such as the way in which prosocial behaviour was measured, both studies suggest that existing differences in social preferences and abilities (i.e., mentalising, SVO) can have a detrimental effect on the neural representations of value following acute stress. Establishing how these changes in neural representations of value impact behaviour following acute stress is a challenge for future work.”

Concerning the power calculation, we have acknowledged this as a limitation in the discussion.

“Our power calculation was based on a 2 x 2 design (Group x Recipient), however, several of our key findings involved three-way interactions (e.g. between Group, Recipient and Effort). Thus, future studies should aim to replicate our effects with larger sample sizes to ensure the robustness of these effects.

**Recommendations for the authors**

**Reviewer #1 (Recommendations For The Authors):**
1. The authors employed an integrative approach on inducing acute stress by combining the strengths of MIST and TSST, as shown by a robust stress response in cortisol. However, some concerns regarding the stress manipulation and the effort-based task need to be addressed. The authors justified the order of deployment as necessary to maintain stress responses throughout the scanning period. It is unclear whether and how potential order effects were controlled, and whether the effort-task performance in the front and back of the line might have different effects in a 90-minute experiment.Moreover, the stress manipulation itself involved a complex mental arithmetic task, which might have influenced participants' willingness to exert effort for others in the prosocial task. As shown in Figure 3, the proportion of participants working decreases as the effort levels increase for both self and other conditions in the stress and control groups. It is thus possible that participants could consider the prosocial task as an opportunity to take a break from the demanding arithmetic task. It would be helpful to present results from the different runs, particularly for the pre and post three runs.

We thank the reviewer for highlighting this potential issue. We have added several analyses to the supplementary analysis to explore potential block effects and fatigue effects. Here we provide a summary of the key findings.

Firstly, we investigated participants’ ratings of the effort levels, which they experienced immediately before and after the study, to investigate potential fatigue effects. We found that following the experiment compared to the before, participants in the stress group rated squeezing to the required effort levels as more physically demanding compared to the control group (p=.037). There were no group differences in how much more effort they reported exerting (p=.824) or how uncomfortable it was (p=.351) compared to before the experiment. Thus, overall the stress group found it more physically demanding to squeeze to the effort levels following the experiment. Crucially, however, increases in how physically demanding participants found it to squeeze to the required effort levels were not correlated with the number of effortful choices in the Self and Other condition in either group (all Ps >0.4). This suggests that whilst stressed participants rated squeezing to the required effort level as more physically demanding following the task relative to before, this was not related to how often participants exerted effort for self or other rewards.

Secondly, we investigated potential block effects. We repeated the mixed effects logistic regression reported in the manuscript but included the interaction between the factors Group, Recipient and Block (1:6) in the model. Although both groups showed a decline in the number of effortful choices during the experiment, the two-way interaction between Group and Block (p=.188) nor the three-way interaction between Group, Recipient and Block were significant (p=.138). This shows that whilst there was a decline in the number of effortful choices throughout the experiment, this was not more pronounced in the stress group, nor was it more pronounced in the stress group for self relative to other effortful choices compared to the control group. Additionally, the key three-way interaction between Group, Recipient and Block was unaffected when controlling for potential block effects. We now also plot the data by block in the supplementary materials (Figure S3).

Please see the section in the Supplementary Material and a summary of these analyses also appears in the manuscript in the Results section

“We conducted additional analyses to rule out the influence of potential fatigue and block effects (see Fatigue and block effects in the Supplementary Materials). In short, the stress group rated squeezing to the required effort level as more physically demanding immediately after the experiment compared to before, which was not seen in the control group (Figure S2). However, this was not related to the number of effortful choices for self or other rewards (Table S2). Moreover, when we conducted the same mixed effects logistic regression on participants’ choices but also included the interaction between Group, Recipient and Block, there was no significant three-way interaction between these factors, nor a significant two-way interaction between Group and Block (Figure S3). Additionally, the three-way interaction between Group, Recipient and Effort was unaffected when controlling for potential block effects (Type III Wald test χ2[4]=22.06, P<0.001). Thus, whilst the stress group rated squeezing to the required effort level as more physically demanding following the experiment, this was not related to the number of effortful choices (for self or other) and the effects of Block on effortful choices (for self or other) did not differ between the group. Thus, changes in how physically demanding participants rated squeezing to the effort levels did not influence decisions to exert effort.”

1. It would be useful to know whether the authors controlled for factors such as familiarity or gender among participants that might influence their choices on the task. If participants were able to interact or observe each other, it is possible that social dynamics played a role in their behavior, which could confound the interpretation of their results. It would be beneficial if the authors could provide further information on how the task was administered and whether any social cues were present.For the experimental design, although salivary samples and subjective pressure were measured, did the authors measure participants' subjective ratings of other negative emotions?

Participants did not have the chance to see or interact with the participants in the “other” condition. Participants were told at the start of the experiment that they would be earning money for the next participant in the study, called Thomas. Thus, as all participants were men, the name of the participants was gender matched. Moreover, as they did not see or interact with the next participant, familiarity was controlled across participants.

We have now added a section p. 8 to clarify this:

“As all participants were men, the name of the next participant was gender matched (all participants were told he was called Thomas; see Methods). Moreover, as participants did not see or interact with the next participant, familiarity was controlled across participants.”

We have now added a plot to the supplementary materials (Figure S4) showing the changes in the ratings of the emotions. Apart from the emotions anxious and disgusted, all other emotions (calm, happy, bad, sad, surprised, angry) showed a significant sample timepoint (1:8) by group (stress, control) interaction, thus mirroring the results for the perceived stress ratings. We now refer to this figure in the manuscript on p. 8:

“for changes in other emotions during the experiment please see Figure S4”

1. Regarding the data analysis section, the authors' analysis is careful overall and the results about SVO are interesting. It would be interesting to know if baseline SVO was similar across both stress and control groups, and if there were any differences in SVO among participants with more individualistic or selfish tendencies. Regarding Figure 4, it would be helpful if the authors clarified whether the vertical coordinate "prosocial choices" is a combination of the five levels of effort or if it is specific to one level. Additionally, it would be useful to explore whether there is a correlation between SVO and prosocial choices and whether effort level could be used as a covariate to control for potential confounding effects. These suggestions could improve the clarity and strength of their contributions.

There were no differences in SVO angle between the control group and stress group (p=.956). There was also a significant correlation between SVO angle and the proportion of prosocial choices across the whole sample. This has now been reported in the manuscript on p. 13:

“There were no existing differences in SVO angle between the groups (control group mean = 19.33, SD = 8.67; stress group mean = 19.23, SD=8.14; p=0.956). We found that across the whole sample – independent of the stress manipulation – there was a significant correlation between SVO angle and the proportion of prosocial choices (r=0.225, P=0.032). So, as expected, those with a more prosocial SVO angle showed a higher proportion of prosocial choices in the task.

To clarify, the variable “% prosocial choices” is a combination of all the five effort levels. In other words, we took the total number of prosocial choices (‘work’ for other) across all effort levels relative to the total number of effortful choices. We have now clarified this in the manuscript on p. 13. As this was a combination of all effort levels (and reward levels), it was not possible to include effort level as a covariate.

“This measure combined all reward and effort levels.”

1. It is noteworthy that in the dACC, an effect was observed with regard to the interaction between perceived stress and SVO angle. Considering this observation, another suggestion would be for the authors to include visualization in Figure 4 to present the results of this interaction. This could help readers better comprehend the findings and provide a clearer representation of the results.

We have now updated Figure 4 so that it has three panels showing the behavioural and neural results concerning SVO angle as well as the relationship between SVO angle and activation to SVself and SVother in the dACC.

1. It would be helpful for readers if the authors could label all statistical plots with appropriate statistical values, effect sizes, and their respective significance levels. By doing so, readers would be able to quickly identify major findings of this study and gauge the degree of significance associated with each plot. The authors should consider including such information in their statistical plots to enhance the comprehensibility of the study results.

We have added statistical values (e.g., beta estimates), including indicators of significance to the plots.

1. The authors selected ROIs based on previous work on stress-related and effort-based decision making (i.e., AI and dACC). While other brain regions may also play a role in decision making and social cognition, the authors could choose to focus on these specific ROIs due to their relevance to the experimental question and hypotheses of this study such as prosocial, mentalizing and subjective values.

We agree that several other ROIs may have also been of interest. However, we decided to restrict our analysis to the dACC and the AI as these two ROIs were the focus of a previous study using the same prosocial effort paradigm (Lockwood et al. 2022) and multiple studies suggest these regions are sensitive to stress effects.

1. The authors chose to use one sample t-test with AUC as a covariate to examine brain activations across all participants regardless of their stress or control condition. This approach could identify brain regions that are associated with perceived stress. However, the authors didn't conduct a simple two sample t-test between stress and control groups since their research question and hypotheses focused on the neurocomputational mechanisms underlying prosocial decision-making during stress. Regarding the different stages of decision-making, such as offer, force, and outcome, the authors did not conduct specific analyses for each stage. Instead, they used the computational model to estimate the subjective value of each option at each stage, which allowed them to examine the neural correlates of different value-related parameters across the entire decision-making process. However, it would be interesting to examine the role of different stages as well.

Our design matrix modelled three events during each trial: the offer, force, and outcome phase (as per Lockwood et al. 2022). However, our hypotheses and research question for the effects of acute stress concerned the offer phase, i.e. when participants were deciding whether to exert effort or not (work vs. rest). Therefore, we decided to limit our reporting to this event. We have clarified this on p. 32 in the Methods:

“Our hypotheses and research questions concerning the effects of acute stress concerned the offer phase, i.e., when participants were deciding whether to exert effort or not (work vs. rest). Therefore, we limited our reporting to this event.”

1. The authors' findings pertaining to individual differences are intriguing, particularly for individuals with selfish tendencies to exhibit lower pro-social tendencies under stress. Additionally, group variations in effortful behavior related to benfitting others, relative to oneself, are more evident at lower effort levels rather than higher ones. The authors could dedicate more space in the discussion section to discuss the potential mechanisms involved and address the absence of pertinent theoretical support.

We have now extended the discussion to further outline potential mechanisms. Broadly, we interpret our findings in terms of compromised executive functioning under acute stress: “downregulation of the brain’s ‘executive control network’ (Hermans et al., 2014)”. In the original submission, we focused on changes in inhibition and shifts to habitual/automatic processing. We have now expanded this to include a section on cognitive flexibility (see below). Note that changes in executive functioning have been widely reported following stress (see Shields et al., 2016 for a meta-analyses). However, which specific executive functions influenced our observed changes in prosocial behaviour is an exciting avenue for future work.

We have added this section on p. 20-21 concerning cognitive flexibility:

“The dlPFC has also been implicated in cognitive flexibility under acute stress. For example, Kalia et al. (2018) used functional near infrared spectroscopy to show that reduced cognitive flexibility under stress was related to changes in activation in the dlPFC in men. In our study, participants in the control group were more likely to exert effort for self rewards compared to other rewards at higher, but not at lower, levels of effort. Whilst participants in the stress group favoured exerting effort for self rewards at every effort level (Figure 3). This consistent preference for self rewards compared to other rewards at all effort level suggests that stressed participants did not adapt their social behaviour in response to changing contextual information. This supports multiple studies showing reduced cognitive flexibility under stress (Goldfarb et al., 2017; Kalia et al., 2018; Raio et al., 2017; Shields et al., 2016). An exciting avenue for future work is to test whether individual differences in executive functions, such as inhibition and cognitive flexibility, predict changes in social behaviour following acute stress. This would be analogous to the finding in non-social domains, where greater working memory capacity protects against stress-induced changes in learning (Otto et al., 2013).

**Reviewer #2 (Recommendations For The Authors):**
The manuscript suggests that the stress group made more selfish responses than the control group at lower, but not higher, levels of effort (as shown in Figure 3). I recommend that Figure 3, showing these data, be modified for clarity. Currently, data for the between-subjects comparison (Control and Stress groups) are linked by a dashed line. This linkage (at least in my mind) connotes that these data points are from the same people at different times. In fact, the within-subjects data are not linked by a line, but are noted by different colored symbols. Please reconsider how these data are presented.

We have redrawn Figure 3. For each effort level, the self vs. other manipulation is shown on the x axis and the two groups (Control vs. Stress) are shown by black and grey lines. For each group, the lines are connected to show that the Self vs. Other manipulation is a within-subject manipulation.